# Enhancing Document Image Retrieval in Education: Leveraging Ensemble-Based Document Image Retrieval Systems for Improved Precision

Yehia Ibrahim Alzoubi [1] , Ahmet Ercan Topcu [2,*] and Erdem Ozdemir [3]

1   College of Business Administration, American University of the Middle East, Egaila 54200, Kuwait; yehia.alzoubi@aum.edu.kw
2   College of Engineering and Technology, American University of the Middle East, Egaila 54200, Kuwait
3   Booking Holdings Inc., 1000 BP Amsterdam, The Netherlands; ozdemircs@gmail.com
*   Correspondence: ahmet.topcu@aum.edu.kw

**Abstract:** Document image retrieval (DIR) systems simplify access to digital data within printed documents by capturing images. These systems act as bridges between print and digital realms, with demand in organizations handling both formats. In education, students use DIR to access online materials, clarify topics, and find solutions in printed textbooks by photographing content with their phones. DIR excels in handling complex figures and formulas. We propose using ensembles of DIR systems instead of single-feature models to enhance DIR's efficacy. We introduce "Vote-Based DIR" and "The Strong Decision-Based DIR". These ensembles combine various techniques, like optical code reading, spatial analysis, and image features, improving document retrieval. Our study, using a dataset of university exam preparation materials, shows that ensemble DIR systems outperform individual ones, promising better accuracy and efficiency in digitizing printed content, which is especially beneficial in education.

**Keywords:** image retrievals; ensemble; education; data model

## 1. Introduction

The rise in mobile phone usage for capturing digital images and the affordability of digital storage systems have shifted the focus towards visual searches instead of traditional text-based searches [1]. This shift has garnered significant interest from the academic community, leading to extensive research in general image retrieval and object recognition. However, there has been a noticeable gap in research regarding DIR with extensive datasets [2]. DIR systems find relevance when printed documents coexist with the need for supplementary digital information. For instance, mobile phones can capture images of printed documents, which can be queried to retrieve specific details. This includes determining the workflow for which the printed copy was created or identifying related documents, offering a versatile approach to access and augment published content with digital information. These systems are also crucial in education, where printed textbooks remain prevalent [3]. For example, students can use their mobile phones to capture sections, exercises, or figures they find challenging or wish to learn more about. A DIR system can then analyze these images, deduce the specific content causing difficulty, and provide additional support. This support might include online tutorials, opportunities for peer discussions, or details about problem solutions and related topics. This approach enables students to access study materials instantly without relying on textual searches. It encourages them to engage in in-depth discussions on subjects they find challenging or want to explore further.

Traditionally, when students encounter difficulties or require additional information about the topics or solutions presented in these books, they turn to their teachers or peers

for assistance. However, due to time constraints, teachers often need help to address every student's query, and the quality of responses may vary. This scenario often leads to the repetition of the same problems being discussed by different teachers with different students across the country, resulting in an inefficiency [4]. We have developed a DIR system for use within a mobile application to address this challenge. The context for this application is the university entrance exams in the Turkish education system, where thousands of students widely use a standardized set of preparation books. These books contain similar problem sets and learning materials. Students can now use their mobile phones to capture images of their questions or problems. The DIR system then processes these images to check if similar queries have been asked and answered previously. The system provides an instant solution or additional information if a match is found. If not, the question is added to a pool for other teachers and students to address, reducing redundancy and promoting more efficient knowledge sharing. This innovative approach enhances the accessibility of educational resources and fosters collaborative learning among students.

During our research, we have observed that relying on a single perspective, such as textual data, global image features, or spatial relations within document images, in a DIR system falls short of achieving the desired results. Each of these single-perspective DIR systems comes with its strengths and weaknesses. To address these limitations effectively, we propose the development of an ensemble of retrieval methods that consider various characteristics of both query and data images. These retrievals then rank the results based on weighted votes. We have designed retrievals with varying levels of effectiveness, termed weak and strong retrievals, depending on their capacity to search extensive document datasets. These retrievals utilize different sets of document image characteristics, such as local structural relationships within images, textual information, run-length histograms, or features extracted in proximity to words. This research represents a notable contribution to both technical and domain-specific knowledge, detailed as follows:

- While many articles have investigated the DIR system, this work stands out as the first to introduce the concept of creating an ensemble of retrievals to achieve high mean average precision. It also provides a framework for developing such ensembles, pushing the boundaries of DIR research. This approach overcomes the limitations of single-perspective DIR systems by combining multiple retrieval methods that consider various characteristics of both query and data images. This ensemble approach leads to more accurate and efficient DIR. This approach assigns higher weights to retrievals with a proven track record of effectiveness, ensuring that the most relevant results are prioritized.
- The design of both weak retrieval and strong retrieval, in this study, is innovative and draws inspiration from various fields, including DIR, image retrieval, and information retrieval. These novel retrieval methods contribute to the advancement of the field. The weak and strong retrievals are categorized based on their ability to handle extensive document datasets. Weak retrievals are computationally efficient and suitable for initial retrieval stages, while strong retrievals offer higher precision for more refined retrieval tasks.
- A particularly distinctive aspect of this research is its application of DIR to the field of education. This pioneering effort bridges the gap between printed textbooks and digital information, offering a fresh perspective on knowledge dissemination in the educational domain.
- Another contribution of this paper is the introduction of two ensemble DIR methods: "Vote-Based DIR" and "The Strong Decision-Based DIR". It also demonstrated that ensemble DIR systems outperform individual ones, promising better accuracy and efficiency in digitizing printed content.

The remaining sections of the manuscript are structured as follows: Section 2 provides an overview of the background problem and reviews pertinent literature in the field. Section 3 discusses the research methodology employed in this study, elucidating the approach taken for investigation. Section 4 presents the research findings, offering insights

and results gleaned from the study. Section 5 engages in an in-depth discussion of the findings, exploring their implications for future research and addressing the current study's limitations. Finally, Section 6 concludes the paper.

## 2. Background and Related Literature

### 2.1. Query Image

Utilizing DIR systems enables individuals to locate digital data such as pertinent documents or supplementary information. DIR systems serve as a conduit connecting printed materials to the digital realm. The demand for DIR systems is particularly pronounced in governmental organizations and businesses where documents exist in both printed and digital formats, necessitating the conversion of printed documents into digital content. In this research, we illustrate the application of DIR systems in education. These DIR systems analyze images captured by students using their mobile phones as input and then search a database of document images to discern the specific query or request made by the student. Particularly in educational settings, DIR systems prove highly beneficial when dealing with complex formulas and intricate figures, which are challenging to locate through conventional text-based searches.

Figure 1 displays a sample set of query images. It is worth noting that the document image dataset contains images resembling these query images. These query images are integrated into the dataset once they have been answered or augmented with additional information. They are typically captured using various mobile phones equipped with different camera features, resulting in varying perspectives and lighting conditions. Some of these images may also include extraneous background details or irrelevant content.

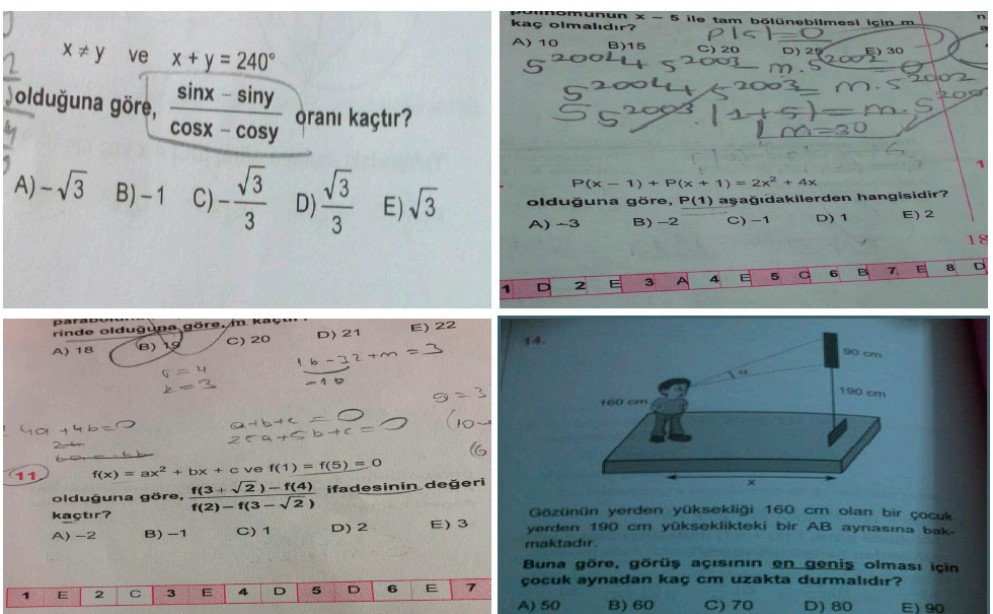

**Figure 1.** Examples of query images.

Additionally, it is common for these images to exhibit imperfections such as scratches. A proficient DIR system must show resilience in response to these image challenges and be able to prioritize relevant images over irrelevant ones. In essence, it should excel in discriminating between images based on their content, regardless of variations in image quality, perspective, and unwanted elements.

### 2.2. Related Literature

A crucial approach in DIR involves leveraging textual content within the document images. The optical code reader (OCR) stands out as the prevalent technique for extracting textual information from document images [5]. As OCR programs have significantly im-

proved accuracy and processing speed, they have become valuable tools for extracting text from document images. Nevertheless, their utility could be more constrained because they are susceptible to issues related to image noise, resolution, and language fields [6]. Furthermore, the realm of information retrieval for textual data has been extensively explored, resulting in the development of numerous text-oriented search engines [7]. AlRahhal et al. [8] introduced a multilanguage framework centered around text transformers. This framework consists of two transformer encoders designed to learn representations specific to different modalities. The initial encoder is a language encoder responsible for creating language representation features derived from textual descriptions. The second encoder is a vision encoder, tasked with extracting visual features from the associated image. In addition to these established methods, OCR-free techniques rely on extracted features from segmented words, such as measurements like word length in pixels or word shape, as well as similar features extracted from segmented characters within the document image [9]. However, it is important to note that all these approaches necessitate the segmentation of words and characters, a process that becomes challenging when characters or words overlap with one another [4].

Alternative studies have explored the use of relationships between high-level components, such as layout, within document images to calculate the similarity between them [10]. Unlike methods reliant on textual information, these approaches do not depend on textual content but may encounter challenges when document layouts are similar [7]. It is worth noting that these studies also necessitate the segmentation of components, which can be a complex task when performed unsupervised [11]. Moreover, there is the potential for different documents to exhibit highly similar layouts, as is the case with problems found in textbooks [12].

More recently, researchers have explored approaches centered on visual information to extract feature vectors, and index structures have been developed to expedite the search process [2,13]. These index data structures employed in these studies excel at swiftly identifying document images that closely match the query image, with only a slight trade-off in terms of a small margin of error [14]. Typically, this margin of error is deemed negligible for fast search operations [15]. Based on the representation of image features, these algorithms can be categorized into two main groups. In the first group, a single feature vector is computed for the entire document image, representing the whole image [6]. Furthermore, some studies adopt graph-based representations to capture the structure of document images and compute graph-matching similarity metrics for ranking similarities [16]. These techniques prove beneficial when the number of comparisons is relatively tiny, as rankings based on pairwise comparisons tend to be less scalable for large datasets [10].

In one approach detailed in [6], run-length histograms were computed at various grid levels to represent the image for retrieval and classification purposes. On the other hand, in studies like [12,17–19], multiple structural features were considered, including the percentages of text and non-text elements (such as graphics, images, tables, and rulings), column structures, relative font sizes, and content area density. In the second group of methods, the focus shifts to detecting local regions within the document images, and feature vectors are computed for these regions [20]. These sets of feature vectors are either aggregated or used individually to represent a single image. For instance, Ref. [18] employed logo and seal detection, which can be particularly beneficial when document images contain illustrations like logos or figures. In another example, Ref. [7] utilized multiscale smearing techniques to locate text and image blocks, while [21] employed affine invariant hashing for local region identification. These diverse approaches demonstrate the evolving landscape of image feature extraction for document retrieval and classification tasks.

In [4], an XY tree deploying index structures for large-scale searches was introduced, and a comparison between locally sensitive hashing and brute force methods was presented. For DIR systems that rely on local regions to represent images, inverted index data structures are commonly employed [21]. Additionally, there is a growing trend of utilizing deep learning approaches in several recent studies [1,3,22–25], reflecting the ongoing evolution

and incorporation of advanced techniques in DIR systems. In this study, our approach is built on the previous literature and distinguishes itself by employing an ensemble of DIR systems rather than creating a single DIR system. This choice is because individual DIR systems relying solely on a single image feature generally fall short in comprehensively representing all aspects of document images and are susceptible to errors under specific conditions. We can mitigate errors from individual plans by seeking consensus across multiple individual DIR systems.

## 3. Materials and Methods

Our research categorizes document search services into two distinct groups: weak DIR systems and robust DIR systems. Within the vulnerable group, we include DIR systems that can swiftly locate the most similar images in extensive document image datasets in less than a second. However, the rankings of these images may be approximate. On the other hand, strong DIR systems employ precise but computationally demanding techniques to calculate pairwise similarities between query and data images. Moreover, strong DIR systems do not exhibit the same scalability as their weak counterparts.

Our study incorporates three different weak DIR systems alongside one strong DIR system. The results generated by these vulnerable DIR systems are subsequently channeled into the strong DIR system. This strong DIR system serves as either the ultimate decision-maker for identifying the most relevant document images for a given query or acts as a prominent voter, with its weight typically surpassing that of the weak DIR systems. This ensemble approach ensures a balanced trade-off between computational efficiency and accuracy in DIR.

The first weak DIR system, "Key-Based DIR", operates by extracting visual features from local regions, specifically contours, within the document image. These visual features are then grouped into clusters. Subsequently, keys are generated based on the spatial relationships between these local regions. The collection of these keys serves as the representation of the document image. Regarding indexing, the key-based DIR employs an inverted index to tally the occurrences of the query keys within the document image dataset. The second weak learner, "OCR-Based DIR", uses OCR to extract text information from the document images. To mitigate potential OCR errors, shingling is applied to the extracted text. This approach enhances the accuracy and robustness of textual information extraction from document images. For textual representation, we employ TFIDF-weighted vectors, and an inverted index structure is utilized to efficiently compute the similarity of textual data across document images and perform ranking.

The other weak DIR system, "Global Image Features-Based DIR", focuses on extracting run-length features from document images. Instead of concentrating on certain document parts, documents are represented and compared in the context of DIR, utilizing features that capture the qualities of the complete image. Features of a whole document's form, texture patterns, and general color distribution are examples of global image features. During the retrieval process, these characteristics are utilized to gauge how similar or dissimilar certain documents are. They also give an all-encompassing depiction of the document picture. Thus, Global Image Features-Based DIR uses features that consider the document picture as a whole to facilitate the efficient and successful retrieval of pertinent documents based on their general visual attributes.

We employ the Fast Library for Approximate Nearest Neighbors (FLANN) library [15]. The FLANN library is a versatile tool used in computer science applications that involve large datasets and the need to efficiently find approximate nearest neighbors. Our proposed robust DIR system, "Word-Based DIR", leverages maximally stable extremal region (MSER)-based detectors to identify local regions within document images [26]. MSER is a computer vision technique that is useful for tasks like object recognition and picture segmentation since it recognizes stable and distinguishable areas in images. Its capacity to manage changes in scale, intensity, and noise in real-world pictures is what makes it so strong. We then extract Speeded-Up Robust Features (SURF) features over these identified regions [27].

SURF is a feature detection and description method that is used for tasks including object recognition, registration, and picture stitching. SURF is made to be more computationally efficient without sacrificing its ability to adapt to changes in light, rotation, and scale. The similarity between two document images is calculated by matching local regions and their features using the random sample consensus (RANSAC) algorithm [28]. RANSAC is an iterative method for robust parameters that are estimated from a set of observed data including outliers that is often used in computational geometry and computer vision. When a model needs to be fitted to data that may contain noise, inaccurate measurements, or outliers, the approach is quite helpful. In the forthcoming sections, we will delve into the specific details of image acquisition and the algorithms utilized within our DIR system.

### 3.1. Image Acquisition

As part of our preprocessing procedure, we initiate the process by binarizing each document or query image. We explored various binarization techniques, including those detailed by Wolf and Jolion [29], FSP [30], and binarization [31]. In our initial assessments, Sauvola's technique demonstrated the fastest processing speed but needed to yield satisfactory binarization results. FSP, on the other hand, exhibited the best binarization performance but was relatively slow in execution. Consequently, we decided to employ the Wolf–Jolion technique for binarizing gray-level document images, balancing processing speed and binarization quality.

### 3.2. Key-Based Document Image Retrieval

In this section, within the Key-Based DIR system, our process begins by employing a Canny edge detector [32] to compute edges within the document image. Following edge computation, we proceed to extract contours. Subsequently, we eliminate contours whose area falls below a predefined minimum contour area threshold. This elimination step proves valuable in reducing noise, which can commonly arise during the binarization process or originate from imperfections in printed paper. We apply a minimum bounding box to each contour found in the gray-level image for feature extraction over these retained contours. Within this bounding box, we resize the image, resulting in an "L $\times$ L" image patch, from which we derive an $L^2$-dimensional feature vector. This process allows us to capture essential visual characteristics from the contours for subsequent analysis and comparison.

The obtained feature vectors are subsequently assigned to C clusters. These clusters are computed using the same set of features over the entire document image dataset. Once the feature vectors are assigned to their respective cluster labels, we process each contour within the document image. We identify the closest contours for each contour and arrange them in order based on their spatial proximity to the target contour. We then concatenate these sorted labels to construct a key. A collection of such keys ultimately represents each document image. Both the document images and the query image are transformed into numerical vectors. These vectors are populated with term frequency–inverse document frequency (TF-IDF) values corresponding to the keys. In text mining and information retrieval, the TF-IDF numerical statistic is used to assess a term's significance in a document for a group of documents [33]. This method takes into account both the frequency of a phrase inside a single document and its rarity over the whole document collection to assist in conveying the relevance of a term within a corpus. We utilize cosine similarity to assess the similarity between the query vector and the document image vector. For efficient processing of large document datasets, we implement TF-IDF and an inverted index using the open-source text retrieval tool SOLR [34]. Apache SOLR, which is an open-source corporate search platform, is a popular text retrieval tool. It is a component of the broader Apache Lucene project, which is a Java-written, feature-rich, fast text search engine library. Building strong and scalable search apps is made possible by the search platform offered by SOLR [34]. This approach ensures scalability in handling extensive collections of document images.

In the context of key-based DIR, the goal is to have each label correspond ideally to a character or a frequently occurring non-textual shape. Importantly, these labels' local arrangement and alignment within the key are preserved. It is worth noting that this approach is not limited solely to characters found in textual content; it can also effectively represent common shapes such as integral signs, triangles, or circles. This flexibility enables the system to capture a wide range of visual elements, enhancing its capacity to identify and retrieve various content types within document images.

### 3.3. OCR-Based Document Image Retrieval

In OCR-based DIR, we employ Tesseract OCR [35] to extract textual information from the document image. Typically, OCRs may not be perfect and can misclassify visually similar characters. To mitigate the impact of OCR imperfections, we adopt a technique where we create d-length shingles of the text. These shingles represent the set of all sequences comprising "d" characters within the text [36]. Once we have extracted shingled text from both the query and the document image dataset, we apply TF-IDF-based weighting and utilize cosine similarity to rank the documents. This approach helps achieve more accurate textual information retrieval while accommodating OCR-related challenges.

### 3.4. Global Image Features-Based DIR

Run-length histograms are widely employed as global image features in document image analysis [6]. The concept of run length refers to the length of a sequence of pixels with the same value in a specified direction. A run-length histogram for an image represents the distribution of these run lengths within the image. Our study considers four distinct directions: horizontal, vertical, diagonal, and anti-diagonal. To quantify the run lengths, we use a logarithmic scale for quantization, which is structured as follows: [1], [2], [3–4], [5–8], [9–16], [17–32], [33–64], [65–128], [129–]. In the context of a binary image, run lengths can be computed separately for both the background and foreground. This implies that the resulting run-length histogram will consist of 18 values in total (9 for the background and 9 for the foreground). To ensure a standardized comparison, these values are typically $L_1$ normalized. This normalization process ensures that the histogram values are scaled appropriately, making them suitable for various analytical purposes, including image feature extraction and similarity assessment.

### 3.5. Word-Based Document Image Retrieval

In this algorithm, we employ a method similar to MSER [26] to locate words within a digital document image. In the case of a high-quality document image, each contour typically corresponds to an individual character. However, as we apply a Gaussian filter to the image, we observe changes in the white spaces between characters. Some characters may start to overlap, leading to contours encompassing more than one character.

Continuing this process iteratively by repeatedly applying the Gaussian filter and computing contours, we eventually reach a point where the contours encapsulate entire words. These word-level contours are stable because the white spaces between words are generally more significant than those between individual characters. To formalize this process, let $\alpha_t$ represent the number of contours at iteration t, and let $\alpha_0$ denote the initial number of contours in the original image. We apply the Gaussian filter and compute the number of contours in successive iterations until the ratio of $\alpha_t$ to $\alpha_t - 1$ exceeds a predefined threshold value, denoted as "L". This threshold helps us identify when the contours have coalesced into stable word-level representations.

Once the number of contours has stabilized, we assume each contour corresponds to a word within the document image. Subsequently, we place a circle around each word and compute the SURF features [27] over these circular regions. The RANSAC algorithm matches these SURF features across different digital image fields [28]. After feature matching, the image similarity is computed based on these matches. It is important to note that this technique is categorized as "strong" within our DIR system. This classification

is because it necessitates the mapping of words between image documents for every pair of query and data images. Furthermore, it is worth mentioning that this approach may need to be better suited for handling a large number of data images due to its computational demands.

*3.6. Ensemble of DIR Systems*

Within our ensemble of retrievals, we introduce two distinct techniques: "Vote-Based DIR" (VBDIR) and "The Strong Decision-Based DIR" (SDBDIR). In SDBDIR, we define $R_i$ as an ordered list of K results, sorted by their similarity to a query image Q, for each of the weak DIR systems denoted as DIR$_i$. We then apply an OR (logical "or") operation to combine these lists $R_i$ for $i = 1 \ldots N$, where $N$ represents the number of weak DIR systems. This operation results in a global candidate set R, the union of all individual lists: $R = R_1 \cup R_2 \cup \ldots \cup R_{|N|}$. The global candidate set R is subsequently provided to the strong DIR system, the final decision-maker. The strong DIR system ranks the documents within the set R based on its criteria and computation. Ultimately, the ranked list produced by the strong DIR system is returned as the final result of the retrieval process.

In the VBDIR approach, both each weak DIR system and the strong DIR system DIR$_i$ are assigned a weight ($w_i$). This $w_i$ reflects the significance of $DIR_i$ in determining the final ranking of relevant documents. Each weak DIR system produces a list of the K most relevant document images denoted as $R_i = \{d_{i1}, d_{i2}, \ldots, d_{iK}\}$ for a given query $Q$, along with their respective similarity scores $sim_i(d_{ik})$ *for* $k \in \{1 \ldots k\}$. The union of the documents from all weak DIR systems is represented as $R = R_1 \cup R_2 \cup \ldots R_{|N|}$, and we determine the membership of each document ($d_j$) in the list $R_i$ using the indicator function $I(d_j, R_i)$, as shown in Equation (1). In essence, VBDIR combines the outputs of multiple weak and strong DIR systems, taking into account their respective weights, to create a unified ranking of relevant documents based on the votes and preferences of each system.

$$I\left(d_j, R_i\right) = \begin{cases} 1 & \text{if } d_j \in R_i \\ 0 & \text{if } d_j \notin R_i \end{cases} \tag{1}$$

$$\pi_j = \sum_{i \in weak} I\left(d_j, S_i\right) \cdot w_i \cdot \frac{sim_i\left(d_j\right)}{sim_i(d_i 1)} \tag{2}$$

$$\pi'_j = \pi_j + \sum_{i \in strong} w_i \cdot \frac{sim_i\left(d_j\right)}{sim_i(d_i 1)} \tag{3}$$

The initial scores ($\pi_j$) for each document $d_j$ are calculated as the sum of weights, which is determined based on the membership function $I(d_j, R_i)$, the $w_i$ assigned to the DIR system, and the ratio of $sim_i(d_j)$ to $sim_i(d_{i1})$. Equation (2) represents the formally expressed calculation. In this equation, $\pi_j$ represents the initial score for document $d_j$. The summation $\Sigma$ is performed over all the weak DIR systems and the strong DIR system. For each system, $i$, the $w_i$, the membership functions $I(d_j, R_i)$, and the similarity ratio $sim_i(d_j)/sim_i(d_{i1})$ are considered in the computation. This scoring mechanism combines various factors to determine the initial relevance score for each document, considering the preferences of different DIR systems and their respective importance weights.

Once the initial assignments of $\pi_j$ for each document have been computed, the similarity of each document is updated based on the scores provided by the strong DIR systems. This update process involves considering the outputs and assessments of the strong DIR systems to refine the document similarity scores, potentially altering their rankings or relevance based on the additional insights and criteria provided by the strong DIR systems. In the final rank, the documents are sorted based on the updated scores $\pi'_j$, as shown in Equation (3). Ideally, strong DIR systems should carry higher weights than weak DIR systems in this ranking process. This approach is designed to mitigate errors that may occur within the strong DIR systems. For example, if all weak retrievals consistently rank a particular document as the most relevant to query Q, but the strong DIR system does not assign it the highest relevance based on its specific criteria, VBDIR, guided by the assigned weights, may prioritize the consensus of the weak retrievals over the judgment of the strong

DIR system. This allows for a balanced consideration of multiple retrieval sources, helping to improve the overall accuracy of the document ranking.

## 4. Experiments and Results

### 4.1. Dataset

For the university exam preparation dataset used in our study, we initially collected approximately 1 million images through a mobile application named Tosbik [37]. However, we conducted a sub-sampling process for this research, selecting 40,000 images related explicitly to mathematics courses. Additionally, we incorporated 100 query images uploaded by students into our system. These query images were intentionally chosen to be challenging. They encompassed a variety of conditions, including different smartphone brands, varying lighting conditions, diverse perspective views, and the possibility of containing scribbles or irrelevant portions of the page. All of these query images were known to have at least one or more relevant documents present within the document image dataset, forming a critical part of our research data.

### 4.2. Experiments

In this study, several parameters needed to be determined. To accomplish this, we divided our query dataset into two sets of equal size: a training set and a test set. We utilized the queries within the training set to identify the optimal parameter values that yield the highest performance based on the mean average precision (MAP) metric. Table 1 outlines the candidate values considered for these parameters and the values ultimately selected for the study. By performing this parameter selection process, we aimed to enhance the overall effectiveness of our approach in terms of retrieval accuracy, as assessed through the MAP metric.

**Table 1.** Parameter, candidates, and selected values used in the study.

| Parameter | Candidates | Selected Value |
|---|---|---|
| C number of clusters in Key-Based | 20, 30, 40, 50, 60 | 50 |
| λ number of labels to generate keys | 3, 4, 5, 6, 7 | 4 |
| L threshold in Word-Based | 0.9, 0.8, 0.7 | 0.8 |
| Minimum contour size | 20, 30, 40, 50 | 20 |

In the vote-based retrieval approach, we assigned weights to each weak retrieval system as follows: $w_i = 1$ for $DIR_i$ belonging to the weak category and $w_i = 2$ for $DIR_i$ belonging to the strong category. It is worth noting that these weights have the potential to be learned from a training dataset, but we left this aspect for consideration in future research. To evaluate our approach, we conducted tests using query images and a dataset of 40,000 document images for each DIR system (OCR, Key-Based, Global Image, and Word-Based). Our evaluation was based on the MAP metric, computed for N = 5, where N represents the number of retrieved records sorted from the most similar to the least similar. This evaluation process allowed us to assess the retrieval performance of our system and compare it across the different DIR methods.

The results of our experiments are as follows: Key-Based DIR, 0.5; OCR-Based DIR, 0.64; Global Image DIR height, 0.74. Notably, the outcomes for the Word-Based DIR system underscore its effectiveness as a strong retrieval system; however, it is important to acknowledge that it may not be scalable for use with large document databases. Despite this, we included the Word-Based DIR system's results in the paper for the sake of completeness and to provide a comprehensive view of our research.

We also calculated the MAP for the SDBDIR and VBDIR ensemble methods. These calculations were performed with various combinations of weak and strong DIR systems, and the results are detailed in Tables 2 and 3. These tables provide insights into the performance of the ensemble methods across different retrieval scenarios, specifically for N = 5.

**Table 2.** VBDIR ensemble retrievals' MAP (SDBDIR).

| OCR + Word-Based | Key + Word-Based |
|:---:|:---:|
| 0.66 | 0.80 |
| **General image + Word-Based** | **OCR + Key + General image + Word-Based** |
| 0.50 | 0.82 |

**Table 3.** SDBDIR ensemble retrievals' MAP (VBDIR).

| OCR + Word-Based | Key + Word-Based |
|:---:|:---:|
| 0.66 | 0.80 |
| **General image + Word-Based** | **OCR + Key + General image + Word-Based** |
| 0.50 | 0.84 |

## 5. Discussion

In this research, we illustrated the application of DIR systems in education. We suggested employing an ensemble of DIR systems rather than creating a single DIR system to achieve this objective. We have devised two general ensemble-based DIR systems, namely VBDIR and SDBDIR. Within these ensembles, we incorporate various DIR systems that utilize diverse techniques. These include an OCR for textual content, spatial analysis of local regions, run-length histogram-based image features, or SURF features around words within the query document image. This approach is designed to retrieve the most relevant document images. Our findings, based on a dataset of university exam preparation books, demonstrate that ensemble DIR systems yield a higher mean average precision in DIR compared to each of the individual DIR systems within the ensemble.

The results (Key-Based DIR: 0.5) reveal that among the weak DIR systems, the Key-Based DIR system delivers the highest performance in terms of the MAP metric. We utilized a widely used non-commercial OCR engine [35], which exhibited decreased performance, especially with blurred and skewed document images. Key-Based DIR outperforms OCR-Based DIR, particularly when dealing with images that contain more graphics than textual information, such as geometry questions. It is worth noting that Key-Based DIR may also be sensitive to blurred images, as its key point localization relies on edge extraction and contour computation. However, it is important to emphasize that the primary focus of this study is to demonstrate the superior performance of ensemble-based DIR systems compared to individual DIR systems. It is possible to enhance DIR systems' robustness against blur, rotation, or skewness by employing different key point detectors and feature extractors.

The Word-Based DIR system demonstrates an ability to handle blurred images due to its focus on locating words rather than individual characters. However, it tends to exhibit poorer performance when document images contain more graphics than textual information. It is important to recognize that these three DIR systems—Key-Based DIR, OCR-Based DIR, and Word-Based DIR—possess distinct characteristics and advantages that make them suitable for different situations and challenges within DIR.

Combining all the DIR systems into an SDBDIR ensemble yields an MAP of 0.82, as shown in Table 2. This demonstrates that leveraging different weak DIR systems with varying characteristics can significantly enhance search performance. Similar findings are observed with VBDIR. It is noteworthy that OCR + Word-Based and Key + Word-Based yield the same results in both VBDIR and SDBDIR, as they involve only a single weak retrieval. Interestingly, when OCR + Key-Based weak retrievals are used with Word-Based strong retrieval, VBDIR outperforms SDBDIR. These results suggest that the errors associated with strong retrievals can be mitigated by considering the decisions made by weak retrievals.

In this investigation, SURF [27] characteristics were also employed to achieve rotational invariance. It is noteworthy that users utilizing the Tosbik [37] program typically

capture photographs in the correct orientation. Although Android phone camera drivers can upload photos in 90, 180, 270, and 360 degrees [38], we efficiently addressed this challenge by rotating the uploaded image four times and organizing the search results accordingly. For applications where users might not adhere to capturing photographs in the correct orientation, delving into rotation-invariant features could be considered for future research endeavors. The image processing segment of this study utilizes C++ to optimize performance and memory utilization. However, potential future research may explore specific structures and the utilization of GPUs to enhance performance.

### 5.1. Future Directions

There are several promising directions for future research in DIR. Firstly, there is room for enhancing the performance of individual weak DIR systems, especially in challenging scenarios like blurred images, skewed documents, or complex layouts. This improvement can be achieved by developing more robust feature extraction methods and key point detection techniques tailored for weak retrievals. Machine learning techniques could also be leveraged to dynamically assign weights to weak DIR systems within ensemble methods. This adaptive weighting strategy could optimize the combination of retrieval approaches based on the specific characteristics of query images and document datasets, potentially leading to more effective results.

Furthermore, our approach can be extended to various domains beyond education. Exploring how DIR systems can benefit fields such as healthcare, legal, or archival settings where document retrieval from printed materials is prevalent presents an exciting avenue for research. Future work can focus on optimizing algorithms and data structures to maintain accuracy while efficiently handling extensive document collections. Another promising area is human interaction with DIR systems, allowing users to provide relevant feedback on retrieved documents. Incorporating user feedback can lead to more personalized and context-aware retrieval results, enhancing the overall user experience. Finally, there is a growing interest in multimodal retrieval, where textual and visual contents within document images are considered for more comprehensive retrieval. Combining OCR-based text retrieval with image content analysis techniques can improve retrieval accuracy and address the diverse information needs of users. These research directions collectively contribute to advancing the field of DIR and expanding its practical applications.

### 5.2. Limitations

The study presents valuable insights into DIR systems but also reveals several limitations that warrant consideration for future research endeavors. Firstly, the sensitivity of OCR-based DIR systems to image quality poses a significant challenge. These systems may struggle with low-quality or distorted document images, affecting overall performance. Future research should prioritize enhancing the robustness of OCR engines to mitigate these issues, ultimately improving system performance across a broader range of image qualities. Another limitation arises from the specificity of the dataset used in this study, primarily consisting of university exam preparation books. While the findings offer valuable contributions, they may not fully capture the diversity of document images encountered in various domains. Future research should extend the evaluation of DIR systems to more extensive and diverse datasets to assess their generalizability and applicability across different document types and domains.

The computational demands of strong DIR systems, particularly those relying on image features like SURF, raise concerns about real-time applications and scalability to large document databases. Addressing computational efficiency remains a critical challenge for future research and will ensure that DIR systems remain practical and accessible for various applications. Lastly, generalization is a critical consideration. The study's focus on specific DIR systems in the context of university exam preparation books should prompt further investigation into the applicability of these systems to a broader range of document types

and domains, ensuring their relevance and effectiveness in various real-world scenarios. Addressing these limitations will pave the way for more robust and versatile DIR systems.

## 6. Conclusions

In this research, we have demonstrated through empirical evidence that combining various DIR systems, each with distinct attributes, surpasses the individual performance of any single DIR system. Moreover, within the ensemble framework, there is potential to create novel weak and strong DIR systems, incorporating diverse features or even alternative index structures, and seamlessly integrate them into the ensemble. Our research findings demonstrate that this ensemble of retrievals significantly outperforms most successful large-scale individual retrieval approaches, as measured by the MAP metric. The retrieval-ensemble approach enhances the accuracy and efficacy of DIR systems, offering a promising solution for handling diverse document images more effectively.

In summary, this work not only introduces groundbreaking concepts in the field of DIR but also showcases the potential for utilizing these advancements in education, marking a significant step forward in the intersection of technology and learning. Looking ahead, one of our future research avenues involves improving the features employed in our retrieval methods to ensure they are rotation-invariant and enhance overall efficiency in speed and memory utilization.

**Author Contributions:** Conceptualization, E.O.; methodology, A.E.T. and E.O.; validation, A.E.T. and E.O.; analysis, Y.I.A.; writing—original draft preparation, E.O. and Y.I.A.; writing—review and editing, Y.I.A. and A.E.T. All authors have read and agreed to the published version of the manuscript.

**Funding:** This research received no external funding.

**Institutional Review Board Statement:** Not applicable.

**Informed Consent Statement:** Not applicable.

**Data Availability Statement:** Data are available on request due to privacy.

**Conflicts of Interest:** Author Erdem Ozdemir was employed by the company Booking Holdings Inc. The remaining authors declare that the research was conducted in the absence of any commercial or financial relationships that could be construed as a potential conflict of interest.

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
