# Peer review of "Enhancing Document Image Retrieval in Education: Leveraging Ensemble-Based Document Image Retrieval Systems for Improved Precision"

_applsci, doi:10.3390/app14020751_

Round 1
Reviewer 1 Report
Comments and Suggestions for Authors
Document Image Retrieval (DIR) systems are crucial in facilitating access to digital data within printed documents. In education, students use DIR to access online materials and find solutions in printed textbooks. However, DIR systems can be improved by using an ensemble of systems, such as "Vote-Based DIR" and "The Strong Decision-Based DIR." These ensembles combine techniques like optical code reading, spatial analysis, and image features, enhancing document retrieval. The study shows that ensemble DIR systems outperform individual ones, proving their effectiveness in digitizing printed content, particularly in education. This research is a pioneering work in the field of Deep Information Retrieval (DIR) by introducing the concept of creating an ensemble of retrievals for high mean average precision. It provides a framework for developing such ensembles, drawing inspiration from various fields, and applies DIR to education, offering a fresh perspective on knowledge dissemination.
This article is well written and organised. I suggest it for publication.
Author Response
Many thanks. Please see the attached document.

Reviewer 2 Report
Comments and Suggestions for Authors
In this paper, the authors propose the development of an ensemble of retrieval methods that consider various characteristics of both query and data images.
I have following comments:
- Many important works published previously for image retrieval have been omitted by the authors for instance a simple search yielded this work:
https://ieeexplore.ieee.org/document/9925582
- The comparison against SOTA models is not done in an appropriate way.
- In terms of contribution, I believe it is not strong enough.
- The authors should conduct also a detailed ablation about the different parameters of the model.
Comments on the Quality of English Language
English writing needs improvement.
Author Response
Many thanks for your valuable feedback. Please see attached file.

Reviewer 3 Report
Comments and Suggestions for Authors
Document Image Retrieval (DIR) systems simplify access to digital data within printed documents by capturing images. These systems act as bridges between print and digital realms, with demand in organizations handling both formats. In education, students use DIR to access online materials, clarify topics, and find solutions in printed textbooks by photographing content with their phones. DIR excels in handling complex figures and formulas. Authors have proposed using an ensemble of DIR systems instead of single-feature models to enhance DIR's efficacy. They introduced "Vote-Based DIR" and "The Strong Decision-Based DIR." These ensembles combine various techniques, like optical code reading, spatial analysis, and image features, improving document retrieval. Their study, using a dataset of university exam preparation materials, shows that ensemble DIR systems outperform individual ones, promising better accuracy and efficiency in digitizing printed content, which is especially beneficial in education. The research presented in the manuscript is of very basic level and is not up-to-the-mark to consider for further review as a journal paper for a SCI-E publication. The novelty of proposed work it too much limited and the results and discussion section is also below average, it is recommended to submit the paper to a conference.
Comments on the Quality of English LanguageMany corrections required.
Author Response

(The authors gave the same response as above.)

Reviewer 4 Report
Comments and Suggestions for Authors
In this research, the authors through empirical evidence that combining various DIR systems, each with distinct attributes, surpasses the individual performance of any single DIR system.
Moreover, within the ensemble framework by the authors , there is potential to create novel weak and strong DIR systems, incorporating diverse features or even alternative index structures, and seamlessly integrate them into the ensemble. Research findings demonstrate that this ensemble of retrievals significantly outperforms most successful large-scale individual retrieval approaches, as measured by the MAP metric. The ensemble of retrievals approach enhances the accuracy and efficacy of DIR systems, offering a promising solution for handling diverse document images more effectively. This aspect no has been improved in the article .
In summary, this work not introduces groundbreaking concepts in the field of DIR in the showcases the potential for utilizing these advancements in education, because the consideration of the a significant step forward in the intersection of technology and learning is not correct. Its possible in the next corrections of this paper involves improving the features employed in others retrieval methods to ensure they are rotation-invariant and enhance overall efficiency in speed and memory utilization.
Author Response

(The authors gave the same response as above.)

Round 2
Reviewer 3 Report
Comments and Suggestions for Authors
The novel contributions of proposed work are too limited, and the research work presented in this manuscript is no up-to-the-mark to consider this for review as a SCI-E indexed paper. authors are encouraged to submit this work in a conference.
Comments on the Quality of English LanguageCorrections required.
Reviewer 4 Report
Comments and Suggestions for Authors
The authors accepted the suggestions and recommendations given in the previous version and adjusted their article to these requirements.